# Diarrhea Is a Hallmark of Inflammation in Pediatric COVID-19

**DOI:** 10.3390/v14122723

**Published:** 2022-12-06

**Authors:** Marco Poeta, Francesco Nunziata, Margherita Del Bene, Francesca Morlino, Alessia Salatto, Sara Maria Scarano, Valentina Cioffi, Michele Amitrano, Eugenia Bruzzese, Alfredo Guarino, Andrea Lo Vecchio

**Affiliations:** 1Department of Translational Medical Science, Section of Pediatrics, University of Naples Federico II, 80131 Naples, Italy; 2Department of Advanced Biomedical Sciences, University of Naples Federico II, 80131 Naples, Italy

**Keywords:** COVID-19, children, diarrhea, gastrointestinal, gut, SARS-CoV-2

## Abstract

Severe acute respiratory syndrome coronavirus 2 (SARS-CoV-2) is a pathogen with enteric tropism. We compared the clinical, biochemical and radiological features of children hospitalized for acute SARS-CoV-2 infection, classified in two groups based on the presence of diarrhea. Logistic regression analyses were used to investigate the variables associated with diarrhea. Overall, 407 children were included in the study (226 males, 55.5%, mean age 3.9 ± 5.0 years), of whom 77 (18.9%) presented with diarrhea, which was mild in most cases. Diarrhea prevalence was higher during the Alpha (23.6%) and Delta waves (21.9%), and in children aged 5–11 y (23.8%). Other gastrointestinal symptoms were most commonly reported in children with diarrhea (*p* < 0.05). Children with diarrhea showed an increased systemic inflammatory state (higher C-reactive protein, procalcitonin and ferritin levels, *p* < 0.005), higher local inflammation as judged by mesenteric fat hyperechogenicity (adjusted Odds Ratio 3.31, 95%CI 1.13–9.70) and a lower chance of previous immunosuppressive state (adjusted Odds Ratio 0.19, 95%CI 0.05–0.70). Diarrhea is a frequent feature of pediatric COVID-19 and is associated with increased systemic inflammation, which is related to the local mesenteric fat inflammatory response, confirming the implication of the gut not only in multisystem inflammatory syndrome but also in the acute phase of the infection.

## 1. Introduction

Children are susceptible to severe acute respiratory syndrome coronavirus 2 (SARS-CoV-2) infection, although the clinical manifestations are generally less severe than in adults. However, a low percentage of pediatric patients may be at risk of severe illness, need of intensive care and death. Children often present with respiratory tract involvement, but a substantial number experience gastrointestinal (GI) symptoms with an overall pediatric prevalence of 40% (range: 13.9–62%) [1]. Diarrhea and abdominal pain are the main GI symptoms in both acute COVID-19 and multisystem inflammatory syndrome (MIS-C), a child-specific, life-threatening, post-infection complication [2].

Similar to other human coronaviruses, SARS-CoV-2 invades not only the respiratory but also the intestinal epithelium. The host receptor angiotensin converting enzyme 2 (ACE-2) is highly expressed in the intestinal epithelia of the ileum and colon, allowing entry and replication of SARS-CoV-2 [3]. In addition, several reports show that the viral RNA shedding in stools is detectable for a longer period than in respiratory specimens [4], but the pathogenesis of COVID-19-associated diarrhea is still under investigation.

Proposed potential enteropathogenic mechanisms, investigated through in vivo and in vitro studies, include: 1. Epithelial damage and mucosal inflammatory response due to enterocytes infection [5,6]; 2. Active transepithelial chloride secretion induced by the spike protein acting as an enterotoxin [7]; 3. Gut dysbiosis due to viral infection [8] or drug-related effects in critically ill patients [9]. 

Since the initial pandemic in December 2019, several variants of SARS-CoV-2 have emerged worldwide, showing an increased transmissibility and a decreased response to available vaccines, causing pandemic waves [10]. Italy, the first European country affected by the pandemic, has thus far experienced four major waves of COVID-19 produced by the spread of the ancestral strain, the Alpha (lineage B.1.1.7), Delta (lineage B.1.617.2) and Omicron (lineage B.1.1.529) variants, which were predominant (>50% of sequenced isolates) during the four waves, respectively. Notably, the accelerated circulation of the Omicron variant led to a rapid increase in hospital admissions for COVID-19 among children and adolescents, driven by higher community transmission, lower antibody neutralization and vaccine effectiveness, and the consequent increased risk of reinfection [11]. 

Currently, data on clinical and laboratory features of children with COVID-19-associated diarrhea are extrapolated from large cohorts of patients, but the risk factors and outcomes of children with diarrhea are not known. Furthermore, it is unclear why only a percentage of patients develop diarrhea. This study aims to compare the clinical, radiological and biochemical features of children (0–17 years) hospitalized for COVID-19, with and without diarrhea. The purpose of this study is to identify possible risk factors, specific clinical patterns, laboratory and radiological features and outcomes of children who develop diarrhea during acute COVID-19.

## 2. Materials and Methods

### 2.1. Study Design and Population

We performed a retrospective cohort study including children with COVID-19 admitted to the Pediatric COVID-19 Referral Center Federico II University Hospital of Naples (Italy) and directly observed by the same group of physicians during the hospital stay. All children aged 0–17 years admitted from April 2020 to May 2022 were considered eligible for the study purpose. SARS-CoV-2 infection was defined by the positivity of validated real-time reverse-transcriptase polymerase-chain-reaction (RT-PCR) assay in at least one respiratory specimen. We excluded children with a diagnosis of MIS-C. We stratified the cases based on the presence of diarrhea defined by the occurrence of three or more loose stools/day [12].

The following information was recorded in an anonymized dataset: demographics, pandemic waves, comorbidities, clinical presentations, laboratory tests, imaging investigations (chest X-ray and abdominal ultrasounds), treatments, time to discharge. GI coinfections were ruled out in stool samples of patients with diarrhea using a multiplex PCR panel (BioFire^®^ FilmArray^®^, Salt Lake City, UT, USA), conventional stool culture and immunoassays detecting rotavirus and adenovirus antigens (ELISA). Pandemic waves were defined according to the local predominance of viral variants: first wave (1 April 2020, to 31 December 2020), second wave (1 January 2021, to 14 June 2021), third wave (15 June 2021, to 13 December 2021), and fourth wave (14 December 2021, to 31 May 2022). Disease severity was defined using the COVID-19 pediatric severity classification recently proposed by Forrest et al. in a large cohort of more than 800,000 children. The severity classification had four levels: asymptomatic, mild (COVID19-related symptoms), moderate (moderately severe COVID19-related health conditions), and severe (unstable COVID-19-related health conditions) [13]. Written informed consent was obtained from the parents of all enrolled children.

### 2.2. Statistical Analysis

Statistical analyses were performed with IBM SPSS Statistics for Windows, Version 26.0 (Armonk, NY, USA: IBM Corp). Continuous variables were reported as mean and SD, or median and IQR, according to their distribution, and compared using a t-test or Mann–Whitney test, as appropriate. Categorical variables expressed as frequencies and percentages were compared using Fisher’s exact test or χ2. Univariable and multivariable logistic regression analyses were used to investigate the variables associated with diarrhea, with risk expressed as crude and adjusted OR (OR and aOR, respectively) with 95% confidence intervals (CI). Multivariable analysis included variables found to have *p* ≤ 0.1 in the univariate analysis.

## 3. Results

### 3.1. Study Population and Diarrhea Prevalence

Overall, 407 children with a diagnosis of acute SARS-CoV-2 infection were included in the study (226 males, 55.5%, mean age 3.9 ± 5.0 years). Eight patients were excluded because SARS-CoV-2 infection was not confirmed by RT-PCR, whereas children with MIS-C diagnosis were excluded. Diarrhea was detected in 77 admitted children, corresponding to an overall prevalence of 18.9% (Figure 1). 

Although in some cases diarrhea was considered a reason for admission, it was often mild with an average duration of 3.5 ± 1.6 days, with a limited incidence of moderate-to-severe dehydration (6.5%). 

Although the difference did not reach statistical significance, diarrhea prevalence tended to be higher during the second (23.6%) and third (21.9%) waves corresponding to the spread of the Alpha and Delta variants, respectively, than during the first (15.7%) and fourth (16.7%) waves, corresponding to the ancestral strain and the Omicron variant, respectively. (Figure 2A). 

The demographic features of enrolled children are summarized in Table 1. Patients with diarrhea did not differ in age, sex, race, vaccination rates or presence of comorbidities compared to those without diarrhea (*p* > 0.05), although diarrhea tended to be more frequent in children aged 5–11 years (23.8%) without reaching statistical significance (Figure 2B). An immunosuppressive state, due to primary immunodeficiency or secondary to chemotherapy or immunosuppressive drugs, was more frequent in children without diarrhea (12.4% vs. 3.8%, *p* < 0.05) (Table 1).

### 3.2. Clinical Features

Table 2 summarizes the clinical features, severity and treatments of enrolled children. 

Fever was the main clinical manifestation in both children with or without diarrhea, with no statistical differences between the groups (68.2% and 74%, respectively, *p* > 0.05). Vomiting, abdominal pain and poor feeding were most commonly reported in children with diarrhea than for those without diarrhea (*p* < 0.001). The incidence of non-GI symptoms did not differ between groups, except for seizures and cough which were more frequent in children without and with diarrhea, respectively (*p* < 0.05). 

The two groups did not show differences in the length of hospital stay and severity distribution (*p* > 0.05), although a higher proportion of patients with moderate or severe COVID-19 were found in the diarrhea group (48.1% and 9% vs. 43% and 6.1%, respectively). One child in the diarrhea group with abdominal fluid collection and sepsis and three patients without diarrhea affected by diabetic ketoacidosis needed intensive care. Deaths were not reported in either group. In addition, more children in the diarrhea group required therapies, mainly parenteral rehydration (*p* < 0.05) and active therapies against diarrhea (probiotics, diosmectite) (*p* < 0.001).

### 3.3. Biochemical and Radiologic Findings

Children with diarrhea showed increased inflammatory markers compared to children without diarrhea, as demonstrated by mean CRP (27.0 ± 65.0 vs. 12.4 ± 25.5 mg/L, *p* < 0.005), procalcitonin (2.8 ± 7.4 vs. 0.5 ± 1.9 ng/mL, *p* < 0.005) and ferritin levels (650 ± 1470 vs. 273 ± 408 ng/mL, *p* < 0.05). Data on inflammatory markers were also confirmed in subgroup analyses excluding either asymptomatic or both asymptomatic and mild-symptomatic cases (Appendix A).

No differences were observed between the two groups in white blood cells, d-dimer and cardiac enzymes (Table 3). An intestinal pathogen in adjunct to SARS-CoV-2 was detected in 7 (9%) children of the diarrhea group, specifically, *Salmonella* species (n = 3), rotavirus (n = 2), adenovirus (n = 1) and *Corynebacterium* species (n = 1). Results based on multiplex PCR were confirmed by bacterial cultures and viral antigens detection. Both rotavirus cases were not vaccinated.

No differences between groups were observed in chest X-ray findings (available for 348 children), with the interstitial pattern being the most common feature in both groups. Sixty-one patients underwent abdominal ultrasound (US) examination. Pathological findings were more frequent in the diarrhea group (70.9% vs. 43.2%, *p* < 0.05).

Figure 3 summarizes the main US findings in children with COVID-19-associated diarrhea. Mesenteric fat hyperechogenicity was the most frequent observed abnormal finding (58.3%), followed by abdominal lymphadenopathy (16.7%), peritoneal effusion (16.7%) and intestinal wall thickening (4.2%).

### 3.4. Univariable and Multivariable Analysis

Children presenting with abdominal pain (aOR 7.36, 95%CI 3.62–14.97), vomiting (aOR 4.37, 95%CI 2.38–8.02) or cough (aOR 1.73, 95%CI 1.05–2.85), as well as those showing pathological abdominal US (aOR 3.06, 95%CI 1.06–8.78) and mesenteric fat inflammation (aOR 3.31, 95%CI 1.13–9.70), had a higher chance of presenting with diarrhea in the univariable analysis (Appendix A). In contrast, diarrhea was less frequently associated with an immunosuppressive state (aOR 0.286, 95%CI 0.086–0.949). Age, sex, race, comorbidities, coinfections, distinct pandemic waves, therapies and pathological X-ray were not associated with the presence of diarrhea in the univariable analysis (*p* > 0.05). 

At multivariable analysis, only the presence of abdominal pain (aOR 5.29, 95%CI 2.37–11.83), vomiting (aOR 2.55, 95%CI 1.27–5.13) and immunosuppression (aOR 0.19, 95%CI 0.05–0.7) retained statistical significance (Table 4).

### 3.5. MIS-C

Seven children admitted to the COVID-19 Referral Center were excluded from the analysis due to MIS-C diagnosis. As shown in Table 5, all MIS-C patients presented with diarrhea and abdominal pain. Inflammatory markers were elevated in all patients, with a cardiac involvement in four. Furthermore, three patients were asymptomatic and four showed diarrhea during the acute phase of the infection.

Diarrhea was watery and severe (10–14 stools/day) in all cases, and stopped after parenteral steroid administration. Six cases had pathological abdominal US, with the fat hyperechogenicity as the most frequent finding. 

Two cases needed admission to intensive care unit: one after abdominal surgery for ab-extrinseco intestinal obstruction by multiple abdominal fluid collections and one for monitoring severe bradycardia due to myocarditis. 

## 4. Discussion

Large clinical studies specifically evaluating COVID-19-associated diarrhea in pediatric population are not available. To the best of our knowledge, the present study is the first that specifically investigated the risk factors and biochemical and radiological characteristics, outcomes and, finally, the management of this specific symptom in pediatric patients hospitalized for SARS-CoV-2 infection. 

In our cohort of children hospitalized for COVID-19, a substantial number presented with GI symptoms, of which diarrhea was the most common. The overall prevalence was slightly increased or within the range of that reported in studies describing the general features of pediatric COVID-19 [14,15,16,17,18]. Indeed, although most attention is still focused on the respiratory symptoms of COVID-19, the number of patients with diarrhea is significant and should not be neglected. 

We found that children with diarrhea did not differ in terms of age, sex, ethnicity and presence of comorbidities compared with other hospitalized children, but why only a percentage of patients develop diarrhea remains unclear. Possibly, differences in the number of ACE-2 receptors in the intestinal tract, individual genetic polymorphisms or differences in the intestinal viral load may predispose a proportion of children to develop GI symptoms compared with others [19,20]. Interestingly, in contrast to a recent study of hospitalized adults [21], an immunosuppressive state was more frequent in children without diarrhea and appears to be a protective factor towards the occurrence of diarrhea in univariable and multivariable analysis. This finding supports the role of inflammation in diarrhea pathogenesis, with its lower risk in children with hampered immunological response due to primary diseases or related to therapies (i.e., steroids, monoclonal antibodies, chemotherapies, immunomodulators or immunosuppressive drugs). 

The frequency of diarrhea tended to be higher during the second and third pandemic waves, corresponding to the local spread of the Alpha and Delta variants. These data are in contrast with a recent study of 1360 Hispanic children reporting a higher prevalence of diarrhea during the Omicron wave compared with previous waves (21.1% vs. 13.3%) [22]. This difference in prevalence may depend on the different settings of the two studies: outpatient children attending the emergency department vs. hospitalized children. Furthermore, in our study, viral variants did not appear to be determinants of GI involvement in univariable analysis, while available in vitro evidences of Omicron variant replication in intestinal cell lines are also discordant [23,24].

Currently, epidemiological and clinical studies indicate that symptoms are generally mild in children and critical cases are rare but are more frequent in adolescents than in younger children [14,15,16,17,18]. In our study, diarrhea was mild and self-limiting in most cases, with a course comparable to that of other viral intestinal infections. However, the number of moderate–severe COVID-19 cases tended to be higher in children with diarrhea. This tendency toward severity is confirmed by the different prevalence of inflammatory markers: children with diarrhea had higher levels of CRP, procalcitonin and ferritin, reflecting a higher systemic inflammatory response [2,25].

Considering the abdominal US findings, mesenteric fat hyperechogenicity was the most common alteration in the whole cohort and was associated with the presence of diarrhea. This specific finding was present also in patients with MIS-C, confirming previous large studies on this severe COVID-19 complication [2,26]. Interestingly, a percentage of children without diarrhea also showed mesenteric fat inflammation, supporting the role of the GI tract in the development of systemic inflammation of COVID-19 induced by the cytokine storm as an active player in the so-called gut–lung axis [27]. In our study, children with diarrhea were at a higher risk of severe systemic inflammation, which reflects the local inflammation of visceral fat induced by viral replication in the intestine. Indeed, mucosal injury leads to alterations in intestinal permeability, increasing the translocation of intestinal luminal contents into mesenteric fat. Pro-inflammatory cytokines produced by adipocytes and macrophages of mesenteric fat trigger systemic inflammation and exacerbate respiratory symptoms by draining through mesenteric lymph [28]. Our in vitro data of the direct enterocyte damage induced by SARS-CoV-2 support its role as an active intestinal pathogen [5,7] and contribute to explaining the increased inflammatory response and, indirectly, even the increased frequency of coughing by children in the diarrhea group.

As expected, children with diarrhea may frequently present with other GI symptoms, particularly vomiting and abdominal pain. The presence of diarrhea did not require specific therapeutic interventions, except for parenteral rehydration and active treatments against diarrhea with probiotics or diosmectite. However, the number of children with moderate-to-severe dehydration was low, and parenteral rehydration was used mainly in cases of poor feeding or to facilitate faster recovery in the isolation ward, differently from MIS-C patients who conversely showed severe dehydration and diarrhea which was only stopped after steroids administration. Differently from critically ill adults [9,29], in our cases, antibiotic administration was not associated with the development of diarrhea. 

The retrospective design is the main limitation of the present study. In order to limit reporting bias, two researchers independently reviewed single patients’ clinical records. Furthermore, this is a monocentric study, which might be an advantage considering that all clinical, radiological and biochemical data were recorded by the same group of physicians, who observed both cohorts of children during their whole hospital stay. Moreover, despite in the initial phase of the pandemic there being a tendency to hospitalize children for a variety of reasons, including only clinical observation due to the novelty of the virus and the very low number of pediatric cases worldwide, our pediatric department has worked as the regional referral center for pediatric COVID-19 since the start of the pandemic, and the flux of hospitalized cases was almost constant across the waves.

Although confirming the mild course of diarrhea in children with COVID-19 similarly to other common viral infections, the present study identifies an association between this symptom and an increased systemic inflammation. Mesenteric fat appears to be the source of pro-inflammatory cytokines, placing the gut as a crucial player in the development of systemic manifestations of COVID-19. Furthermore, diarrhea is a major clinical feature of the post-infection inflammatory complication MIS-C, and our study adds information on the course of children with diarrhea also during acute COVID-19, supporting a direct link between intestinal involvement and systemic inflammation. In particular, the presence of diarrhea, even if mild, is a hallmark of severity in terms of systemic inflammation in both the acute phase and MIS-C. Our findings are in line with a recent study showing a higher risk for developing MIS-C during hospitalization in patients admitted for GI symptoms and positive for SARS-CoV-2 [30]. 

Finally, considering that new variants are likely to continue to emerge, that only a small percentage of children have been vaccinated for COVID-19, and that pandemics have resulted in a number of missed rotavirus vaccinations, we expect an increase in hospitalization rates for acute gastroenteritis due to SARS-CoV-2 and other viruses in the future, imposing a great deal of attention in this area, particularly for children with COVID-19-related GI symptoms, both for acute management and for the subsequent development of MIS-C.

## Figures and Tables

**Figure 1 viruses-14-02723-f001:**
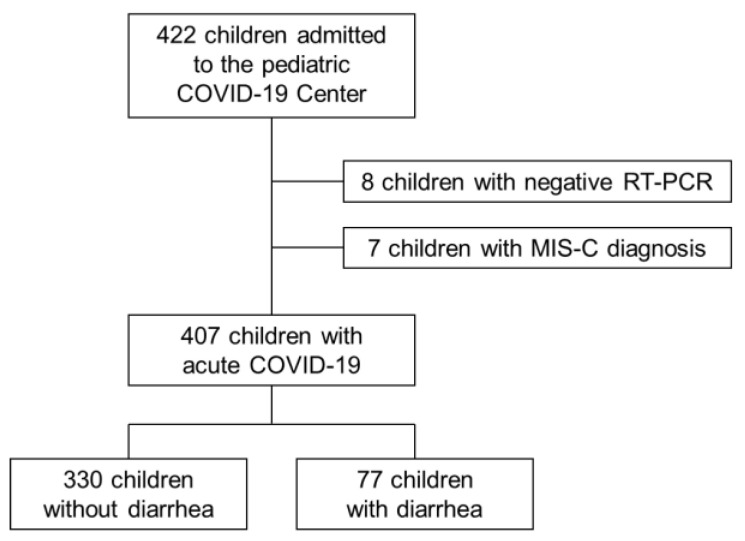
Flow diagram of study population.

**Figure 2 viruses-14-02723-f002:**
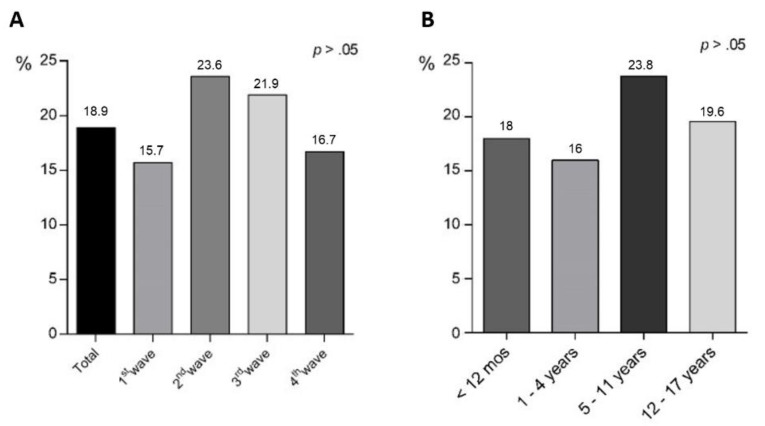
Prevalence of diarrhea in SARS-CoV-2-infected children according to pandemic waves (**A**) and child’s age (**B**). χ2 test was used to compare groups.

**Figure 3 viruses-14-02723-f003:**
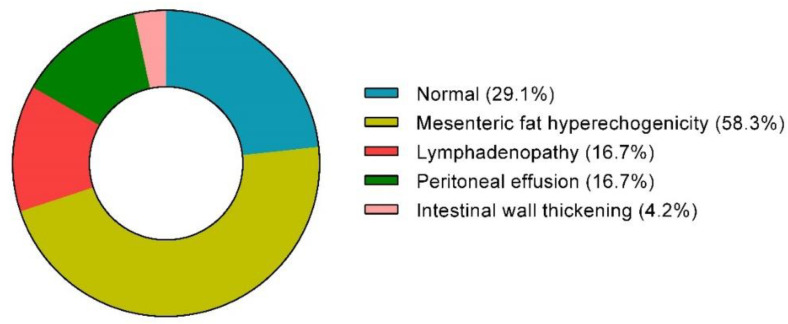
Abdominal ultrasounds’ findings in SARS-CoV-2-infected children presenting with diarrhea.

**Table 1 viruses-14-02723-t001:** Demographic features of enrolled children.

	Overall Population (n = 407)	Children with Diarrhea (n = 77)	Children without Diarrhea (n = 330)	*p*
Mean age, y (SD)	3.9 (5.0)	4.3 (5.2)	3.8 (5.0)	0.525
<12 m, n (%)	177 (43.5)	32 (41.4)	145 (44)	0.704
1–4 y, n (%)	100 (25)	16 (21)	84 (25.4)	0.391
5–11 y, n (%)	84 (20.6)	20 (26)	64 (19.4)	0.199
12–17 y, n (%)	46 (11.3)	9 (11.6)	37 (11.2)	0.905
Male sex, n (%)	226 (55.5)	47 (61)	179 (54.2)	0.330
Non-Caucasian, n (%)	23 (5.7)	4 (5.2)	19 (5.8)	0.923
At least one comorbidity, n (%)	166 (40.8)	29 (37.7)	137 (41.5)	0.536
Cardiovascular, n (%)	25 (6.1)	3 (3.8)	22 (6.7)	0.362
Respiratory, n (%)	28 (6.9)	7 (9.1)	21 (6.4)	0.395
Neurological, n (%)	31 (7.6)	7 (9.1)	24 (7.3)	0.588
Immuno-rheumatological, n (%)	7 (1.7)	1 (1.3)	6 (1.8)	0.752
Onco-hematological, n (%)	43 (10.6)	5 (6.5)	38 (11.5)	0.197
Genetic-metabolic, n (%)	21 (5.2)	5 (6.5)	16 (4.8)	0.557
Hepato-gastroenterological, n (%)	16 (3.9)	4 (5.2)	12 (3.6)	0.526
Nephro-urological, n (%)	8 (2)	0 (0)	8 (2.4)	0.168
Endocrinological, n (%)	17 (4.2)	3 (3.8)	14 (4.2)	0.891
Immunosuppression, n (%)	44 (10.8)	3 (3.8)	41 (12.4)	0.030
COVID-19 vaccination, n (%)	29 (7.8)	6 (7.8)	23 (7.0)	0.164
Period of hospital admission				
1st wave (ancestral strain), n (%)	83 (20.4)	13 (16.9)	70 (21.2)	0.740
2nd wave (Alpha variant), n (%)	89 (21.9)	21 (27.3)	68 (20.6)	0.203
3rd wave (Delta variant), n (%)	73 (17.9)	16 (20.8)	57 (17.3)	0.470
4th wave (Omicron variant), n (%)	162 (39.8)	27 (35.1)	135 (40.9)	0.203

Abbreviations: m, months; y, years.

**Table 2 viruses-14-02723-t002:** Clinical features, severity classification and therapies of enrolled children.

	Overall Population (n = 407)	Children with Diarrhea (n = 77)	Children without Diarrhea (n = 330)	*p*
**Clinical features**				
Fever, n (%)	282 (69.3)	57 (74)	225 (68.2)	0.317
Peak body temperature, mean °C (SD)	38.4 (0.7)	38.6 (0.8)	38.4 (0.7)	0.683
Cough, n (%)	157 (38.6)	38 (49.4)	119 (36.1)	0.031
Nasal discharge, n (%)	84 (20.6)	17 (22.1)	67 (20.3)	0.729
Sore throat, n (%)	24 (5.9)	5 (6.5)	19 (5.8)	0.805
Dyspnea, n (%)	49 (12)	7 (9.1)	42 (12.7)	0.377
Croup, n (%)	6 (1.5)	0 (0)	6 (1.8)	0.233
Vomiting, n (%)	55 (13.5)	24 (31.2)	31 (9.4)	<0.001
Abdominal pain, n (%)	37 (9.1)	21 (27.3)	16 (4.8)	<0.001
Poor feeding, n (%)	51 (12.5)	13 (16.9)	38 (11.5)	<0.001
Asthenia, n (%)	41 (10.1)	10 (13)	31 (9.4)	0.346
Headache, n (%)	20 (4.9)	7 (9.1)	13 (4)	0.060
Seizures, n (%)	34 (8.4)	2 (2.6)	32 (9.7)	0.043
Chest pain, n (%)	11 (2.7)	3 (4)	8 (2.4)	0.473
Skin lesions, n (%)	24 (5.9)	7 (9.1)	17 (5.2)	0.186
Length of hospital stay, mean (SD)	5.9 (6.8)	6.4 (8.3)	5.8 (6.4)	0.578
**Severity classification ***				
Asymptomatic, n (%)	14 (3.4)	0 (0)	14 (4.2)	0.177
Mild, n (%)	187 (45.9)	33 (42.9)	154 (46.7)	0.546
Moderate, n (%)	179 (44)	37 (48.1)	142 (43)	0.424
Severe, n (%)	27 (6.6)	7 (9)	20 (6.1)	0.336
Critical, n (%)	4 (1)	1 (1.3)	3 (0.9)	0.755
Deaths, n (%)	0 (0)	0 (0)	0 (0)	-
**Therapies**				
Antibiotics, n (%)	155 (38.1)	36 (46)	119 (36.1)	0.089
Systemic corticosteroids, n (%)	39 (9.6)	10 (13)	29 (8.8)	0.260
LMWH, n (%)	10 (2.5)	2 (2.6)	8 (2.4)	0.922
Oxygen supplementation, n (%)	23 (5.7)	5 (6.5)	18 (5.5)	0.727
Parenteral rehydration, n (%)	74 (18.2)	20 (26.0)	54 (16.4)	0.048
Probiotics, n (%)	18 (4.4)	18 (23.4)	0 (0)	<0.001
Diosmectite, n (%)	3 (0.7)	3 (3.9)	0 (0)	<0.001

Abbreviations: LMWH, low-molecular-weight heparin; n, number; SD, standard deviation; *, severity classification according to Forrest et al., 2022.

**Table 3 viruses-14-02723-t003:** Biochemical and radiological features of enrolled children.

	Overall Population	Children with Diarrhea	Children without Diarrhea	*p*
**Biochemical parameters**	**(n = 407)**	**(n = 77)**	**(n = 330)**	
CRP, mg/L mean (SD)	15.2 (33.2)	27.0 (65.0)	12.4 (25.5)	0.002
PCT, ng/mL mean (SD)	0.9 (2.9)	2.8 (7.4)	0.5 (1.9)	0.004
Ferritin, ng/mL mean (SD)	344 (608.9)	650 (1470)	273 (408)	0.034
Neutropenia, n (%)	111 (27.3)	16 (20.8)	95 (28.8)	0.147
WBC (10^3^ cells/μL), mean (SD)	8.5 (4.7)	9.1 (5.7)	8.3 (4.5)	0.166
Neutrophils (10^3^ cells/μL), mean (SD)	3.5 (3.6)	4.0 (4.2)	3.4 (3.5)	0.230
Lymphocytes (10^3^ cells/μL), mean (SD)	4.5 (2.6)	4.1 (2.4)	4.6 (2.6)	0.477
Platelet count (10^3^ cells/μL), mean (SD)	294 (131)	316 (159)	289 (125)	0.113
D-Dimer, ng/mL mean (SD)	1573 (4083)	2170 (7124)	1434 (3374)	0.211
CK-MB, ng/mL mean (SD)	1.3 (10.2)	2.3 (1.8)	4.2 (12.2)	0.091
hs-cTn, pg/mL mean (SD)	23.0 (85.4)	8.1 (14.4)	26.5 (102)	0.144
Any coinfections, n (%)	81 (19.9)	20 (25)	61 (18.5)	0.138
**Chest X-ray**	**(n = 348)**	**(n = 67)**	**(n = 281)**	
Normal	107 (30.7)	19 (28.4)	88 (31.3)	0.637
Pathological	241 (69.3)	48 (71.6)	193 (68.7)	0.637
Interstitial	163 (46.8)	32 (47.8)	131 (46.6)	0.866
Lobar/GGO	78 (22.4)	16 (23.9)	62 (22.1)	0.749
**Abdominal ultrasounds**	**(n = 61)**	**(n = 24)**	**(n = 37)**	
Normal	28 (45.9)	7 (29.1)	21 (56.8)	0.035
Pathological	33 (54.1)	17 (70.9)	16 (43.2)	0.035
Abdominal lymphadenopathy	7 (11.5)	4 (16.7)	3 (8.1)	0.306
Mesenteric fat hyperechogenicity	25 (41.0)	14 (58.3)	11 (29.7)	0.026
Intestinal wall thickening	4 (6.6)	1 (4.2)	3 (8.1)	0.544
Peritoneal effusion/fluid collections	7 (11.5)	4 (16.7)	3 (8.1)	0.306

Abbreviations: CK-MB, creatine kinase-myoglobin binding; CRP, C-reactive protein; GGO, ground-glass opacities; hs-cTn, high-sensitivity cardiac troponin; n, number; PCT, procalcitonin; SD, standard deviation; WBC, white blood cells. Normal values of biochemical parameters are listed in Appendix A.

**Table 4 viruses-14-02723-t004:** Factors associated with the occurrence of diarrhea in univariable and multivariable analysis. Variables found to have *p* < 0.10 at univariable analysis (see Appendix A) were included in multivariable analysis.

	Univariable Analysis			Multivariable Analysis		
Factors	aOR	95%CI	*p*	aOR	95%CI	*p*
Vomiting	4.368	2.379–8.020	<0.001	2.549	1.266–5.132	0.009
Abdominal pain	7.359	3.619–14.966	<0.001	5.290	2.366–11.825	<0.001
Cough	1.728	1.048–2.849	0.032	1.563	0.891–2.743	0.119
Seizures	0.248	0.058–1.060	0.060	0.287	0.064–1.296	0.105
Headache	2.438	0.939–6.334	0.067	1.931	0.660–5.652	0.230
Antibiotics administration	0.648	0.393–1.070	0.090	1.472	0.836–2.590	0.180
Pathological abdominal US	3.055	1.063–8.774	0.038	1.802	0.717–4.532	0.211
Mesenteric fat inflammation	3.309	1.129–9.695	0.029	3.109	0.959–10.078	0.059
Immunosuppression	0.286	0.086–0.949	0.041	0.187	0.050–0.700	<0.001

Abbreviations: aOR, adjusted odds ratio; CI, confidence interval; US, ultrasounds.

**Table 5 viruses-14-02723-t005:** Features of children with MIS-C.

	MIS-C Patients (n = 7)
Mean age, y (SD)	9.0 (2.7)
Male sex, n (%)	3 (43)
Caucasian, n (%)	7 (100)
Comorbidities, n (%)	0 (0)
Fever, n (%)	7 (100)
GI involvement, n (%)	7 (100)
Diarrhea, n (%)	7 (100)
Vomiting, n (%)	4 (57)
Abdominal pain, n (%)	7 (100)
Normal chest X-ray, n (%)	7 (100)
Cardiac involvement, n (%)	4 (57)
Abdominal US, n (%)	
Normal	1 (14)
Lymphadenopathy	3 (43)
Mesenteric fat hyperechogenicity	6 (86)
Intestinal wall thickening	2 (29)
Peritoneal effusion/fluid collections	3 (43)
ICU admissions, n (%)	2 (29)
CRP, mg/L mean (SD)	195.4 (140.8)
PCT, ng/mL mean (SD)	10.4 (8.7)
Ferritin, ng/mL mean (SD)	510.3 (346.8)
WBC (10^3^ cells/μL), mean (SD)	14.4 (7.9)
Neutrophils (10^3^ cells/μL), mean (SD)	10.1 (8.4)

Abbreviations: CRP, C-reactive protein; MIS-C, Multisystem Inflammatory Syndrome of Children; PCT, procalcitonin; US, ultrasounds; SD, standard deviation; WBC, white blood cells.

## Data Availability

The datasets generated and/or analyzed during the current study are available from the corresponding author on reasonable request.

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
