# Peer review of "Diarrhea Is a Hallmark of Inflammation in Pediatric COVID-19"

_viruses, 2022, doi:10.3390/v14122723_

Round 1
Reviewer 1 Report
This study by Poeta et al is a retrospective study design with the aim of evaluating inflammatory marker levels in children with acute covid, comparing those with diarrhea to those without. Overall the study can add to the current literature on acute COVID illness in children as pediatric clinical presentation and pathology overall is still limited compared to what we know about adult illness. The authors conclude that inflammatory markers are higher and abdominal imaging is abnormal in children with COVID presenting with diarrhea as compared to those without diarrhea as a symptom and this suggests more severe disease and systemic inflammation in these patients.
There were strengths to this study including a large cohort of 407 children with acute COVID infection included in the analysis. Children had a wide range of severity and symptoms. It also included children admitted across multiple waves (i.e. variants) of the SARS-CoV-2 pandemic ranging from the initial varient in 2020 through the most recent Omicron wave.
There are improvements that can be made particularly regarding the comparisons and statistics to better support the conclusions that inflammatory markers are elevated in the setting of diarrhea and this could signify more severe disease. Below are some major and minor critiques which can improve the study and manuscript.
1. Lines 108 and 121 – the statements that the prevalence of diarrhea was higher in the 2nd and 3rd wave and in 5-11yo group is somewhat misleading as this suggests statistical significance. This should be revised to reflect the qualitative nature of the conclusion as a trend to higher frequency but did not reach significance.
2. Recommend adding the number (n) of subjects per wave in each group (with and without diarrhea). This could be added to table 1, similar to the format for the age ranges for example. Differences in admissions across waves could bias the data in Figure 2 as early in the pandemic there may have been a tendency to admit children with covid for a variety of reasons including observation due to the novelty of the virus and unknowns associated with illness or very few numbers early because of lower number of pediatric infections. If there are significant differences in the total "n" across waves that should be noted.
3. Table 3: Regarding the comparisons of inflammatory lab values – the inclusion of asymptomatic children in the non-diarrheal group or admissions for very mild upper respiratory symptoms or observation of high-risk populations (such as asymptomatic immunocompromised patients) could skew/bias the data towards higher inflammatory markers in the diarrhea group which were all symptomatic and in need of inpatient level of care. Sub-analysis of the inflammatory marker data matched across severity scores and comparing inflammatory markers between the groups in the differing severity classes would strengthen the study. If power is too low across severities then comparing only symptomatic children to each other, i.e. excluding asymptomatic cases or grouping those separately would be a better comparison.
4. Line 163-165 and Table 3: The presence of intestinal pathogens may be pertinent to the clinical presentations but the relevance can’t be evaluated in this cohort as the data is presented – How many children in the non-diarrheal group had stool testing performed? This was likely very few as there would be no reason to test the stool in that group, so the presence of pathogens such as Salmonella sp. and adenovirus may not be related to the SARS-CoV-2 diarrhea as these can be shed for a long time following prior infection. Additionally the 2 rotavirus positives could be vaccine related depending on the age of the child. It is reasonable to state these were identified but the interpretation should be qualified with any limitations if there is lack of this testing or information in the non-diarrheal group and the validity of the statistical analysis for this conclusion is limited unless there is an n for negative stool tests in the comparison group. Also, for both the total coinfection and GI co-infection data – the mode of identification (culture vs. PCR/nucleic acid testing) should be listed in the methods.
5. Line 261 – the statement that there was a "direct correlation" indicated between the presence of GI symptoms an elevated inflammatory markers and severity is inaccurate as there was no statistical difference in severity classes between diarrheal and non-diarrheal groups as shown in Table 2. This sentence should be revised.
6. Lines 303-305 – The statement that close follow-up or trending inflammatory markers would be suggested by this data should be reconsidered in the context of the data presented or additional data to support such as statement. What is the evidence that children with diarrhea in the acute illness stage are at higher risk of MIS-C? The paper that is referenced (reference 30), focused on patients where gastrointestinal symptoms were in the early/presenting stages of PIMS/MIS-C immediately prior to the MIS-C diagnosis (within days) and then were diagnosed with MIS-C during their admission for the GI symptoms. This is not a directly comparable group to this study as this focused on acute infection which usually precedes MIS-C by 2-6 weeks. If the data is available - Did any of the patients in this cohort (with or without diarrhea) go on to develop MIS-C or did any of the MIS-C cases have preceding diarrheal illness with their acute COVID illness (weeks prior)? This statement should be revised to better reflect the data as presented and our current knowledge of pediatric acute COVID and MIS-C pathogenesis. If there were patients that went on to develop MIS-C after their acute admission in either group, this is important and should be mentioned.
7. Authors should comment on vaccination rates as the latter waves (specifically Omicron) could have included some vaccinated children in each group, depending on availability, approval, and recommendations for pediatric COVID vaccination in Italy during this study.
Minor suggestions:
8. Figures/tables/statistics throughout manuscript: Some p values are listed as “.000” – this is confusing and should have either the exact p value listed even if scientific notation is needed for very small p values, or use a lower limit cutoff, eg. <0.0001.
9. Figure 2 – Statistical analysis approach/test used should be described briefly in the legend.
10. Line 139 – This sentence is confusing – seizures and cough are listed as more frequent in children without and with diarrhea. Recommend splitting this into 2 independent interpretations/sentences for clarity rather than using respectively
11. Laboratory standard ranges for the labs should be listed (even if supplemental materials) as different labs may have different cutoffs for what is elevated.
Reviewer 2 Report
I read with interest this work presented by Poeta et al. entitled (Diarrhea is a hallmark of inflammation in pediatric COVID-19). I congratulate the authors for this well-articulated and informative manuscript. Diarrhea has been previously reported as a not rare symptom of patients with COVID-19. The importance of this work is the focus on diarrhea and its relationship with the outcome of COVID-19, and it seems as a good marker of inflammation as was shown by biochemical tests. The manuscript is well introduced, with sound methodology, and clear results and all important parts are well discussed. Yet, some minor comments needed to be answered by the authors to improve this version of the manuscript.
# In abstract: please write the full word followed by the abbreviation when it is mentioned for the first time. (ex. SARS-CoV-2, M, y, ..... etc)
# For the COVID-19 pediatric severity classification the authors referred us to another paper (Forrest et al. 2022), it would be good if the authors write a few sentences about the classification.
# L 285- 286: The authors need to discuss how diarrhea due to COVID-19 was differentiated from antibiotics-associated diarrhea (AAD), as it is not mentioned in the exclusion criteria for diarrhea that developed during hospitalization. Although AAD is rarely reported in children.
# Regarding GI coinfections that were reported in 9% of the included patients. I guess bacterial causes were detected via culture. But how the viral ones were detected? molecular methods? The authors need to discuss that in the results and I guess it is another limitation for this study if rapid tests or serology were used for the detection of other viruses such as Rotavirus, Adenovirus, Norovirus, Astrovirus....,. The seasonality of diarrheagenic viruses was reported in children and this study was conducted over two years period.
# Table CRP is nicely presented as mean and SD and it is already significant, I see there is no clue to be mentioned as a % of high CRP.
If these comments were considered, this manuscript could be benefited from another round of revision.
Best regards
